# The Idiopathic Pulmonary Fibrosis-Associated Single Nucleotide Polymorphism RS35705950 Is Transcribed in a MUC5B Promoter Associated Long Non-Coding RNA (AC061979.1)

**DOI:** 10.3390/ncrna8060083

**Published:** 2022-12-08

**Authors:** Ruxandra Neatu, Ifeanyi Enekwa, Dean J. Thompson, Edward C. Schwalbe, Giorgio Fois, Gina Abdelaal, Stephany Veuger, Manfred Frick, Peter Braubach, Sterghios A. Moschos

**Affiliations:** 1Department of Applied Sciences, Faculty of Health and Life Sciences, Northumbria University, Ellison Building, Newcastle-Upon-Tyne NE1 8ST, UK; 2Translational and Clinical Research Institute, Faculty of Medical Sciences, Newcastle University, Central Parkway, Newcastle-Upon-Tyne NE1 3BZ, UK; 3Institue of General Physiology, University of Ulm, Albert-Einstein-Allee 11, D89081 Ulm, Germany; 4Institute of Pathology, MHH Hannover, 30625 Hannover, Germany

**Keywords:** non-coding RNA, MUC5B, IPF, rs35705950

## Abstract

LncRNAs are involved in regulatory processes in the human genome, including gene expression. The rs35705950 SNP, previously associated with IPF, overlaps with the recently annotated lncRNA AC061979.1, a 1712 nucleotide transcript located within the MUC5B promoter at chromosome 11p15.5. To document the expression pattern of the transcript, we processed 3.9 TBases of publicly available RNA-SEQ data across 27 independent studies involving lung airway epithelial cells. Epithelial lung cells showed expression of this putative pancRNA. The findings were independently validated in cell lines and primary cells. The rs35705950 is found within a conserved region (from fish to primates) within the expressed sequence indicating functional importance. These results implicate the rs35705950-containing AC061979.1 pancRNA as a novel component of the MUC5B expression control minicircuitry.

## 1. Introduction

Human DNA consists of protein-coding regions and non-coding regions. Protein-coding genomic regions are abundantly transcribed, evolutionarily conserved, mutationally sensitive sequences which impact cellular phenotype. These constitute approximately 1% of the human genome [1]. Non-coding regions of DNA, on the other hand, are more complex and can be divided into at least five structural types: (i) binding motifs for regulatory proteins, (ii) non-coding RNAs (ncRNAs), (iii) transposable elements, (iv) highly repetitive DNA—essential in gene regulation and chromosome maintenance, and (v) pseudogenes [2,3]. In 2003, the ENCODE (Encyclopedia of DNA Elements) project was launched to identify and classify functional elements in the human genome including non-coding transcripts. The project continues to grow with the results being made available on Ensembl and UCSC genome browser for both human and mouse [4].

Most of the ncRNAs predicted by ENCODE are expressed at low levels [4]. However, their abundance is not a proxy for their functionality [5]. For example, the predicted lncRNA ENST00000567151 or viability enhancing in lung Cancer transcript (VELUCT) was found at only 0.01 copies per cell. Despite its low copy number, VELUCT expression was reported to be upregulated by 5.2 fold in lung cancer cells, and its knockdown reduces the viability of multiple lung-cancer cell lines by as much as 90% [6].

Recently, lncRNAs (long ncRNAs) have been intensively studied due to their involvement in cancer [7], neurological conditions [8], pulmonary diseases [9], and the regulation of chromosome structure [10]. This research culminated in several published lncRNA databases: NONCODE [11], Lnc2Cancer [12], LncRNADisease [13], and LncRNAdb [14]. LncRNAs can regulate expression via at least two mechanisms: *cis*-acting lncRNA (which regulate expression of adjacent genes) and *trans*-acting lncRNAs (regulating the expression of distant genes on other chromosomes) [15]. Most pancRNAs (promoter associated ncRNAs), to date, have been associated with an increased production of mRNA from the adjacent protein-coding gene, suggesting that pancRNAs might contribute to gene expression regulation. Protein-coding genes that possess pancRNAs also exhibit tri-methylated lysine 4 of histone 3 (H3K4me3) and acetylated lysine 27 of histone histone 3 (H3K27ac), whereas the pancRNA-free genes appear to lack such epigenetic signatures [16]. However, how pancRNAs change the expression of protein-coding genes remains unknown.

Furthermore, evidence is amassing concerning the expression of pancRNAs and the occurrence of epigenetic changes. Thus, typically, when single nucleotide polymorphism (SNPs) appear in such non-coding transcript loci, the associated pancRNA secondary structure is disrupted, affecting expression patterns and impacting upon the function [17]. Whilst expression changes in high-copy-number lncRNA are easy to determine by routine RNA-SEQ, the effect of SNPs resulting in small changes in lncRNA expression levels is harder to study.

The G/T rs35705950 SNP found in the promoter of mucin 5B (MUC5B) on chromosome 11p15.5 [18,19] has one of the highest (~40%) [20] and most reproducible associations with idiopathic pulmonary fibrosis (IPF) across white, hispanic, and Asian populations [21,22,23,24,25,26,27,28,29,30,31,32,33,34,35,36,37], with homozygous mutants exhibiting a higher risk of developing the disease [38] and higher mortality [39]. The polymorphism is implicated in the elevated transcription and translation of MUC5B in both healthy and diseased individuals [40]. This is evidenced via episomal expression of luciferase driven by TT or GG MUC5B promoters cloned from IPF patients in A549 alveolar epithelial cells [41]. Since MUC5B is one of the largest proteins encoded in the human genome, excessive expression is proposed to lead to elevated endoplasmatic reticulum (ER) stress [42] through MUC5B protein recycling and the unfolded protein response, increasing cell sensitivity to exogenous insults and pro-apoptotic phenotypes. This is exacerbated in alveolar lung epithelia, where MUC5B aberrant mRNA expression is elevated but MUC5B protein production is not normally observed. Presently, the polymorphism is thought to a) disrupt a 25 CpG motif differentially methylated region which is, counterintuitively, hypermethylated in IPF, and b) enhance the binding of the transcription factor Forkhead Box Protein A2 (FOXA2), 32 bp downstream of the SNP, as evidenced by chromatin immunoprecipitation [18]. Given the distal effect of the SNP to the FOXA2 binding site and the emerging role of pancRNA in transcription regulation, we sought to determine whether an lncRNA transcript might be implicated in MUC5B expression and its transcriptional dysregulation in the context of the rs35705950 SNP.

To this end, we analysed publicly deposited RNA-SEQ datasets. However, most pipelines for novel transcript discovery are focused on small RNA populations or certain RNA species [43], and RNA-SEQ workflows typically involve polyadenylated transcript enrichment. This creates a classical signal-to-noise-ratio detection problem where selective signal acquisition and amplification during sequencing-library preparation may reduce non-polyadenylated transcript read frequencies to levels typically ascribed to background noise. Inspired by the application of very long base interferometry in expanding observation dynamic range beyond standard signal-to-noise-ratio limitations through signal integration from multiple sources operating similar data acquisition protocols [44], we applied composite analysis of third-party RNA-SEQ datasets to reveal the existence of such technically occluded transcripts. Overall, we describe a novel and simple computational method for performing such de-novo lncRNA transcript searches by aggregating data from diverse input sources, and focusing analytical efforts on the RNA-SEQ-verse to specific genomic regions of interest.

## 2. Results

### 2.1. Rediscovery of the Non-Coding Transcript AC061979.1 in the Promoter Region of the MUC5B Gene

To identify a putative non-coding transcript in a “dark” intergenic region on the p-terminus of chromosome 11 on the human genome in the context of lung epithelia, we manually collected and interrogated a total of 3.9 TBases of publicly available RNA-SEQ data involving epithelial lung cells (see Appendix A) to generate a summative transcriptional signal of the AC061979.1 locus (see Figure 1A). Concomitantly, the GENCODE [45] release 32 (GRCh38) described the putative transcript AC061979.1 (chromosome 11:1,218,530–1,220,242) mapping to the same region (see Figure 2). Interestingly, this novel lncRNA is reported to be subject to splicing, with rs35705950 mapping to the second nucleotide of exon 2 and, therefore, possibly altering AC061979.1 splicing; however, no evidence of splicing was immediately apparent through our analysis (see Figure 1) and no rs35705950-containing RNA-SEQ data from lung epithelia were found among the surveyed studies, with the exception of three donor samples across two independent studies (SRP096589/GSE93526 and SRP102483/GSE97036) where the expression of AC061979.1 was documented (see Appendix A).

Manual inspection of each dataset indicated that almost all samples representing lung epithelial cell lines or primary lung epithelial cells showed evidence of expression in the locus. Of particular interest, however, was a dataset (SRP082973) that isolated only basal cells from the epithelium. Thus, within the same study, we compared basal cell and the epithelium (mix of basal, ciliated, columnar and secretory cells) RNA expression [46]. Interestingly, whilst reads from epithelial cell extracts mapped liberally to the AC061979.1 locus, sporadic alignments of only a couple of reads were detected among basal-cell RNA (Figure 1). Given that basal cells are a sub-type of human-airway epithelial cells not involved in mucous production, which act as stem cells for the other sub-types (ciliated, columnar and secretory cells) [47], these results potentially show the activation of ncRNA AC061979.1 after cell differentiation. Analysis of data from IPF studies (see Figure 1C) involving lung tissue (GSE52463 or SRP033095), primary cells (GSE116086 or SRP151008), and at single-cell level (GSE124685 or SRP175341) demonstrated only limited coverage of the locus in line with the low expression level indicated elsewhere, and confirmed the absence of AC061979.1 expression in fibroblasts. Taken together, these results suggest that expression at AC061979.1 is detectable in lung epithelia irrespective of the biological origin of the data or the precise sequencing protocol used, minimising the risk of batch-associated effects.

To determine the evolutionary importance of the DNA sequence harbouring the rs35705950 polymorphism, the human reference genome (GRCh38.p13) was aligned against nine vertebrate genome seuqences: six mammals (Rhesus monkey, baboon, marmoset, pig, sheep, rat, mouse), one fish (zebrafish) and one bird (chicken). High similarity was observed in exonic regions across primates, with phylogenetically distant mammals showing conservation only at the 5${}^{\prime }$ end of the second exon (see Figure 3A). This region appears to harbour at least three conserved loci, including a FOXA2 binding motif and four SMAD binding motifs, two of which reside in the putative intron (see Appendix A) and are found across mammalian species (see Figure 3B).

Although AC061979.1 transcript abundance in other species is limited by the lack of RNA-SEQ datasets to interrogate, taken together these results indicate a functional significance for this pancRNA, with the G/T rs35705950 SNP possibly being involved in differential splicing of AC061979.1.

### 2.2. MUC5B pancRNA Expression Validation

To independently validate the expression of AC061979.1, we first designed probe hydrolysis RT-qPCR assays for the putative spliced variant and holotranscript. These assays exhibited amplification efficiencies of 128.8% and 90.6% when tested against serial dilutions of a spliced AC061979.1 geneblock or A549 cell DNA extracts, respectively. Next, we obtained RNA extracts from adenocarcinoma human alveolar type-II epithelial cells (A549 cells) and cystic-fibrosis bronchial epithelial cells (CFBE41o-) representing alveolar and bronchial epithelial cells, respectively. To account for the potential impact of contact inhibition effects on AC061979.1 expression, total RNA was extracted at low (<30%) and high (>70%) confluence, and expression of the two AC061979.1 variants was assessed against 18S rRNA and MUC5B, across serial dilutions of total RNA. These analyses indicated that only the AC061979.1 full transcript was detectable, albeit at a very low copy number (see Figure 4). Thus, at 50 ng of RNA input per RT-PCR reaction, in A549 cells the paRNA ΔCt to 18S was 22.44 (±5.94) at high confluence vs 24.12 (±3.16) at low confluence, whereas in CFBE41o- the ΔCt was 21.92 (±5.53) at high confluence vs 28.59 (±0.81) at low confluence (*n* = 3). Of note is that where RNA extraction resulted in higher Ct values for 18S, the capacity to detect the pancRNA transcript was lost as concentrations dropped below the assay limit of detection, justifying the very high load of RNA template in the RT-PCR reactions.

The same assays were perfomed with undifferentiated (basal) and ALI differentiated HAEpCs. The basal cells were cultivated to a confluence of 40–50% before extracting total RNA. Of the two AC061979.1 transcripts, only the full variant was detectable (36.19 ± 0.29), whereas the spliced variant was not detectable. Interestingly, MUC5B levels were below the assay detection limit; 18S was detected at a Ct of 10.63 (±0.14). ALI-differentiated HAEpCs in control conditions showed similar values to the basal cells (36.84 ± 1.38), whereas in IL-13-stimulated cells both AC061979.1 variants were below the detection limit. In differentiated epithelia, Cts for 18S were low for both controls and IL-13-treated cells (7.97 ± 0.06 and 8.53 ± 0.08, respectively). Similarly, to the basal cells, MUC5B was not detectable. Efforts to define the 5${}^{\prime }$ and 3${}^{\prime }$ ends of the transcript by RLM-RACE failed to produce sequencing-grade amplicons, probably due to the low expression level of the transcript.

To ascertain the relevance of the FOXA2 binding motif in exon 2 of AC061979.1, we examined FOXA2 expression levels in A549 cells and HAEpCs. This analysis indicated no statistically significant difference between A549 cells at different confluence levels (*p* = 0.1000), but a 10.6 ± 2.77-fold reduction in FOXA2 levels after IL-13 stimulation (*p* < 0.0001) consistent with the loss of AC061979.1 (see Appendix A).

## 3. Discussion

MUC5B dysregulation presently appears to be mechanistically involved in the development of the underlying pathology, particularly in the context of the IPF-associated SNP rs35705950. It contributes to mucus overproduction and expression in the alveolar microenvironment, leading to micro-injuries to alveolar epithelium and, across the lifetime of a carrier, excessive cell death and fibrosis [48]. Whilst in one study the polymorphism was found in 51% of the patients with IPF, but in only 23% of the control group [18], it is unclear at present if the onset of disease among rs35705950 positive controls is a matter of time, lifestyle, or additional genetic variability. However, the strong association and high incidence rate of the polymorphism in IPF make a compelling case for lifestyle management and preventative chronic or genome modifying treatments targeting MUC5B expression repression and IPF, such as small interfering RNA, antisense or genome/prime editing. The four putative SMAD binding sites within the AC061979.1 locus, four of which reside in the proposed intron, suggest complex interplay between SMAD as an inducer and FOXA2 as a repressor of MUC5B.

Helling et al. (2017) reported a binding motif for FOXA2, located 32 bp downstream of rs35705950, which overlaps with the second putative exon of the pancRNA AC061979.1, as reported in GENCODE v32 (see Figure 2). The protein-coding gene for FOXA2 originates on chromosome 20, p11.21, between the lncRNA LINC00261 and LNCNEF. Whilst LINC00261 (a.k.a. DEANR1 [49], FALCOR [50], and LCAL62 [51]) is widely studied for its role in non-small cell lung cancer, no study exists on LNCNEF to date. LINC00261 is an endoderm-associated lncRNA which recruits SMAD2/3 to induce the expression of FOXA2 [49,51]. FOXA2 transcription factor is known to have a role in lung development and homeostasis [52], MUC5B expression and IPF [53]; however, research in this area is limited.

Our RNA-SEQ-verse survey did not return an explicit splicing signal in line with RNA-SEQ observations associated with high copy number RNAs (see Figure 5); However, if the proposed splicing event is confirmed, the location of the SNP raises the possibility that aberrant AC061979.1 splicing might be occurring in the context of the rs35705950 SNP. In turn, improper AC061979.1 splicing could be driving aberrant biochemistry on the locus such as the FOXA2 association demonstrated by Helling et al. (2017). Such a finding would introduce the additional option of a splice-correcting treatment in preventing the onset of IPF among rs35705950 carriers. Importantly, this oligonucleotide therapeutic modality has been approved for clinical use in Duchenne’s muscular dystrophies (eteplirsen) and spinal muscular atrophy (nusinersen) without the need for drug-delivery solutions that otherwise plague efficacious oligonucleotide therapies for the lung [54].

LncRNAs can form complex biological systems by binding to other RNA molecules, regulatory proteins, or DNA. FENDRR is an lncRNA expressed in the nascent lateral mesoderm, in the promoter of Forkhead Box F1 (FOXF1), where it forms a triple helix with double-stranded DNA and increases the occupancy of the Polycomb repressive complex 2 (PRC2) at this site. Rescue experiments on FENDRR-knockdown cells wherein a construct expressing the lncRNA was placed randomly in the genome showed its biological role and that the transcript acts in *trans* [55]. Similarly, LINC00261-null cells were rescued by viruses expressing FOXA2, in the transcriptional activation of FOXA2, which is upstream of LINC00261 [49]. It is, thus, possible that the mechanism behind MUC5B regulation involves an assembly between the pancRNA AC061979.1 and other regulatory proteins or transcripts interacting with the promoter region of MUC5B acting in *cis* or in *trans*, including the competitive binding of FOXA2 or SMAD2/3. Although Helling et al. (2017) did not assess the importance of SMAD2/3 in MUC5B expression, Feldman et al. (2019) showed that phosphorylated SMAD levels are low in mucosecretory cells, and the inflammatory TGF-beta-dependendent SMAD signaling inhibition enhanced mucin expression, as well as goblet-cell metaplasia and hyperplasia, supporting a role for SMAD proteins in MUC5B expression regulation [56]. Interactions with SMAD2/3 in the promoter of MUC5B and AC061979.1 are indeed possible due to the presence of the canonical SMAD binding element (SBE) CAGAC within the intronic region of the pancRNA, and the newly described GGC(GC)(CG) motif also known as 5GC SBE [57] within the first exon of AC061979.1 [56]. Moreover, SMAD2/3 does not necessarily need to occupy either of these SBEs on chromatin, because SMAD2/3 does not occupy the SBEs located within the LINC00261 gene [49]: instead, it interacts with LINC00261 directly at least under some experimental conditions [50]. LINC00261 is, therefore, an example of cis-acting ncRNA, whereas other lncRNAs such as EMT-associated lncRNA induced by TGFbeta1 (ELIT-1) act in trans to bind SMAD to SMAD binding elements (SBEs) such as the CAGAC box [58]. Disruption of this proposed SMAD2/3, AC061979.1, and possibly the FOXA2 ribonucleoprotein and chromatin interaction network at the MUC5B promoter by rs35705950, for example, due to aberrant splicing, could explain MUC5B overexpression in IPF, given the pivotal role of SMAD proteins in resolving goblet-cell metaplasia and hyperplasia in inflammatory pulmonary disease [56].

To date, the FOXA2 binding site 32 bp downstream of rs35705950 has been shown to bind FOXA2 in episomal reporter systems but not by genome editing or CHIP-SEQ [18]. Our own genome-editing efforts with three separate single-guide RNAs to introduce the rs35705950 G/T transversion at Chr11:1,219,991 in A549 cells in support of CHIP-SEQ, RIP-SEQ, and proteomic experiments to resolve the MUC5B transcriptional complex, have so far proven to irreparably affect cell viability or fail in generating any detectable editing either by T7-EI or sequencing assays. Furthermore, no verified G/T or T/T lung epithelial cell line is currently available to support such mechanistic studies. As lncRNA–protein interaction is a hot research topic, recent studies have focused on developing computational methods for predicting these complex networks [59,60,61,62]. It is, thus, anticipated that with increasing understanding of lncRNA biology and characterisation of lncRNA structures and families, additional insights into AC061979.1 function might be obtained.

In this study, we developed a simple-to-use method for the targeted mining of the RNA-SEQ dataverse for lncRNA transcripts irrespective of their polyadenylation status. Our method is achievable on a public server in Galaxy (galaxyproject.org) with an extensive easy-to-follow guide available (see Appendix A). It takes as input Sequence Read Archive (SRA) codes and the output is a .TXT file reporting the depth of coverage per position making end-user memory requirements compatible with standard desktop/laptop computers or even smartphones. However, it can be adapted to run on a cluster without a graphical user interface (GUI). Using this method, we have been able to amass evidence through the analysis of 3.9 TBase of RNA-SEQ data across 27 publications documenting the expression of a novel pancRNA overlapping the IPF-associated rs35705950 SNP implicated in MUC5B overexpression, annotated as AC061979.1 by GENCODE. The results were replicated by qRT-PCR in A549 cells and CFBE41o submerged cultures as well as in pHAECs.

## 4. Materials and Methods

### 4.1. RNA-SEQ Data Processing for Novel ncRNA Detection

To determine the existence of a MUC5B pancRNA, we manually collected publicly deposited RNA-SEQ data from 27 independent studies involving alveolar and bronchial samples from primary human tissue and invitro experiments (see Appendix A). RNA-SEQ reads above Q20 were mapped to the human reference genome GRCh38.p13 using HISAT2 [63]. Mapped reads were filtered with samtools view [64] and only read pairs mapping to chromosome 11, region 1,202,000–1,220,500, were kept. Subsequently, the depth of coverage per base was extracted from all datasets and collapsed. The results were visualised in R Studio (ggplot2). The pipeline can be performed in Galaxy (galaxyproject.org). An extensive step-by-step guide is available as a Appendix A.

### 4.2. Multiple Sequence Alignment

To demonstrate the evolutionary importance of the region overlapping the promoter polymorphism rs35705950, we compared the human ncRNA with nucleotide sequences of 10 other species from fish to primates. Rhesus monkey (*Macaca mulatta*), baboon (*Papio anubis*), white-tufted-ear marmoset *Callithrix jacchus*, pig (*Sus scrofa*), sheep (*Ovis aries*), Norvegian rat (*Rattus norvegicus*), house mouse (*Mus musculus*), chicken (*Gallus gallus*) and zebrafish (*Danio rerio*). Alignments of genome sequences were undertaken using AVID and Shuffle-LAGAN programs implemented through mVISTA (http://genome.lbl.gov/vista/mvista/submit.shtml, accessed on 10 January 2022) [65] with a match criterion of 70% identity over 50bp [66]. All sequences used in analysis are included in Appendix A. Subsequently, we aligned the same genomic sequences with ClustalOmega for nucleotide-by-nucleotide approach.

### 4.3. Cell Culture

A549 cell passages 10–12 were thawed, seeded on t25 flasks at 37 ${}^{{}^{\circ}}$C 5% CO_2_ in DMEM/F12 (1:1) (ThermoFisher Scientific, Cramlington, UK) with 10%FBS, +1% L-Glutamine, 1% Penicillin/Streptomycin (Merck Life Science UK Ltd., Dorset, UK). Cells were cultivated till 50–60% confluency, then split in other t25 flasks until 20–30% confluency was reached (usually 24h). CFBE41o- cells (passage 10–12) were thawed, then seeded on T25 flasks at 37 ${}^{{}^{\circ}}$C, 5% CO2 in MEM (Merck Life science UK limited) with 10% FBS, 1% L-Glutamine, 1% Pen/Strep (Merck Life Science UK Ltd.). Cells were cultivated till 50–60% then seeded into new t25 flasks until 20–30% or 50–60% confluency was reached for subsequent low-confluency or high-confluency total RNA extractions, respectively.

Primary human airway epithelial primary cells (pHAECs) from several donors (*n* = 1 basal cells, *n* = 4 ALI differentiated cells) were isolated from fresh tissues that were obtained during tumor resections or lung transplantation with the full consent of patients (ethics approval: ethics committee Medical School Hannover, project no. 2701-2015).

In addition, pHAECs basal cells (passage 4) were cultivated on T75 Flasks in airway epithelial cell basal medium supplemented with airway epithelial cell growth medium supplement pack and with 5 μg/mL Plasmocin prophylactic, 100 μg/mL Primocin and 10 μg/mL Fungin (all from InvivoGen, Toulouse, France). Trypsinization with Promocell DetachKit (Promocell, Heidelberg, Germany) and RNA etxraction was performed at ~40–50% confluency.

pHAECs basal cell for air liquid interface (passage 2) were expanded as above in T75 flasks till 90% confluency. The cells were than trypsinized and seeded into Transwell filters (6.5 mm diameter, 4 μm pore size, Corning Costar, Kaiserslauten, Germany). Filters, prior to cell seeding, were coated with 100 μL collagen solution (StemCell Technologies, Saint Égrève, France), and left to dry under sterile hood overnight. Subsequently, the filters were exposed to UV light for 30 min and stored at 4 ${}^{{}^{\circ}}$C.

Cells were resuspended in growth medium, and 200 μL containing 4 × 10^4^ cells were added apically to each filter, an additional 600 μL of the medium were added basolaterally. The medium was replaced every 48 h until 100% confluence was reached. Growth medium was then removed from apical side and on the basolateral side it was replaced with ALI differentiation medium ±10 ng/ml IL-13 (IL012; Merck Millipore). Once the ALI interface was established, medium was exchanged every second day till day 25–28 on ALI. At the endpoint of cultivation, RNA extraction was performed directly on the filter.

### 4.4. RNA Extraction

RNA extraction was carried out using the miRNeasy mini kit (Qiagen, Manchester, UK). Briefly, cells were detached by trypsinisation then resuspended in 0.7 mL Qiazol Lysis reagent with subsequent steps according to the supplier’s total RNA extraction protocol.

For pHAECs, cells were detached by trypsinisation then resuspended in 2.1 mL Lysis Solution RL from my-Budget RNA Mini Kit (BioBudget, Krefeld, Germany), RNA isolation was performed following the manufacturer protocol. If not used immediately after lysis, the samples were stored at −80 ${}^{{}^{\circ}}$C. For pHAECs ALI cultures, 100 μL of Lysis Solution RL from the same kit was added to the filters apically and the samples were immediately frozen at −80 ${}^{{}^{\circ}}$C.

### 4.5. DNase Treatment and cDNA Synthesis

Total RNA was DNase treated using the Precision${}^{TM}$ DNase kit Primer Design (Southampton, UK), following the manufacturer’s protocol. cDNA synthesis was carried out using the High Capacity cDNA reverse Transcription Kit (Thermo Fisher scientific (ThermoFisher Scientific) following the manufacturers protocol. A total of 1000 ng of total RNA was loaded into each 20 μL cDNA synthesis reaction.

For pHAEC cells, the cDNA synthesis from the extracted was performed using SuperScript VILO cDNA Synthesis Kit (Thermo Fisher) following the manufacturer protocol. A total of 400 ng of RNA were used for each reaction.

### 4.6. Real-Time Quantitative PCR

Custom primers and probes (Appendix A) were designed using the PrimerQuestTM tool (Integrated DNA Technologies BVBA, Leuven, Belgium) and validated against an AC061979.1 geneblock (Integrated DNA Technologies) corresponding to the predicted spliced transcript. Inventoried predesigned assays for 18S and MUC5B were purchased from Thermo Fisher Scientific, and for FOXA2 from Qiagen. Real-time quantitative PCR was performed in 10 μL reactions containing 5L TaqMan Fast Advanced Master Mix (2×) (ThermoFisher Scientific), 900 nM forward primer, 900 nM reverse primer and 250nM probe per reaction and 1μL template on a StepOnePlusTM real-time PCR system (Thermo Fisher Scientific). After a UNG incubation at 50 ${}^{{}^{\circ}}$C for 2 min, initial denaturation at 95 ${}^{{}^{\circ}}$C for 2 min was followed by 40 cycles of 95 ${}^{{}^{\circ}}$C denaturation for 1 s and 60 ${}^{{}^{\circ}}$C anneal extension for 20 s. Gene expression was calculated according to the delta Ct method [67]. Statistical analyses on gene expression were performed on data expressed as a fold difference to high confluence A549 samples, and control samples in a paired sample fashion for HAEpCs, respectively. GraphPad Prism v.9.4.1 (GraphPad Software. LLC, San Diego, CA, USA) was used for Kolmogorov–Smirnov tests for cell-line results and paired t tests for HAEpC results. For RNA ligase-mediated 5${}^{\prime }$ and 3${}^{\prime }$ rapid amplification of cDNA ends (RLM-RACE), the FirstChoice RLM-RACE kit was used according to the manufacturer’s instructions (Thermo Fisher) using pancRNA gene-specific primers for RT-PCR.

## Figures and Tables

**Figure 1 ncrna-08-00083-f001:**
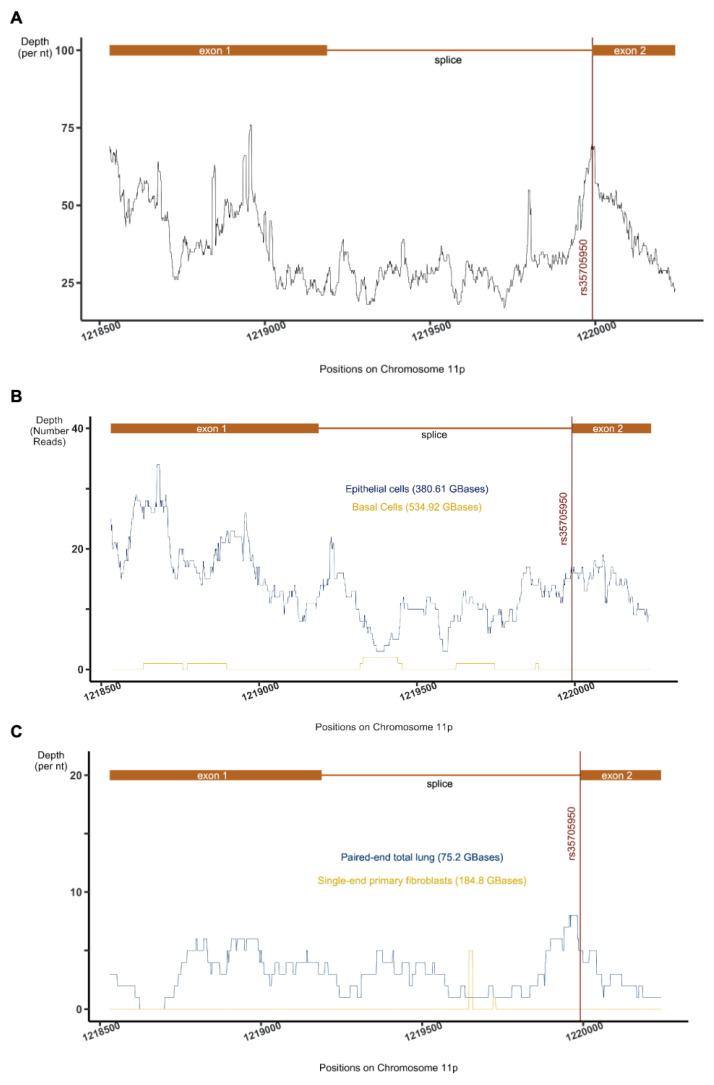
RNA-SEQ data processing results. Depth of coverage spanning chromosome 11: 1,218,530–1,220,242 collected from (**A**) 27 studies (3.9 TBases), (**B**) a single dataset (SRP082973) comparing epithelial to basal cell expression, and (**C**) 3 IPF-related lung-tissue (SRP033095) and fibroblast (SRP151008 and SRP175341) studies. The position of rs35705950 is indicated by a red vertical line and the AC061979.1, primary transcript and spliced exons are indicated in orange.

**Figure 2 ncrna-08-00083-f002:**
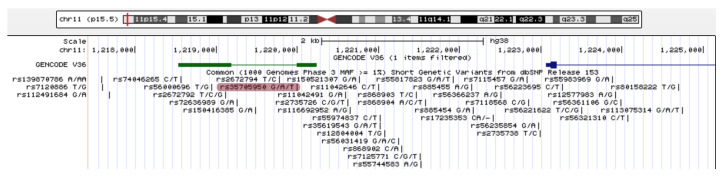
UCSC genome browser Genomic location of the annotated lncRNA AC061979.1. The putative pancRNA AC061979.1 is located on chromosome 11, at 1,218,530–1,220,242—green; the transcription start site of MUC5B—dark blue; thick line—exons; thin line—introns; rs35705950—highlighted in red.

**Figure 3 ncrna-08-00083-f003:**
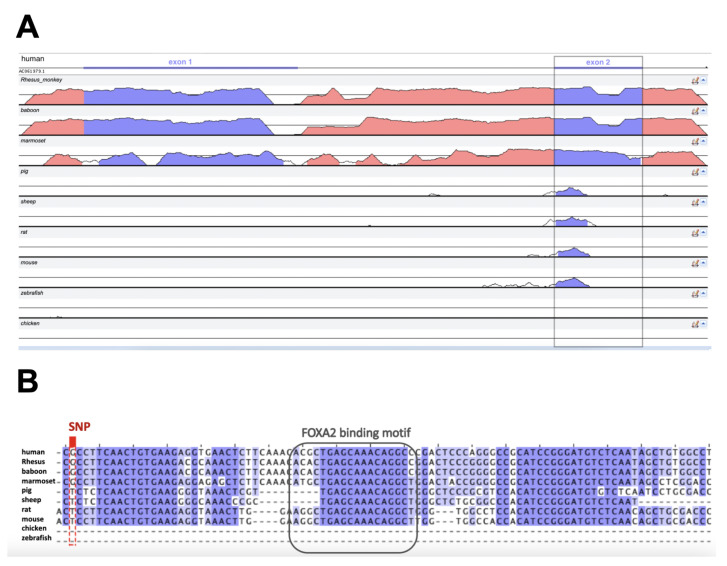
Conservation of the MUC5B-MUC5AC intergenic region across 10 species. (**A**) The genomic sequences were aligned using AVID in mVISTA: global pair-wise alignment between ~2000 nt spanning the human AC061979.1 transcript and the whole intergenic region of the other species (~20,000 nt; Appendix A). Coloured peaks (purple: AC061979.1 exons; pink: intergenic regions) indicate at least 50 bp with 70% similarity. The grey rectangle indicates the conserved exon across mammals. (**B**) Multiple Sequence Alignment by ClustalOmega in Jalview shows that 100 nucleotides downstream of rs35705950 (red rectangle) there are (i) 15–25 bp conserved across mammals (purple shades by nucleotide similarity percentage), (ii) a FOXA2 binding site (grey rectangle), and (iii) a third conserved region approximately 10 nt downstream of the FOXA2 binding site.

**Figure 4 ncrna-08-00083-f004:**
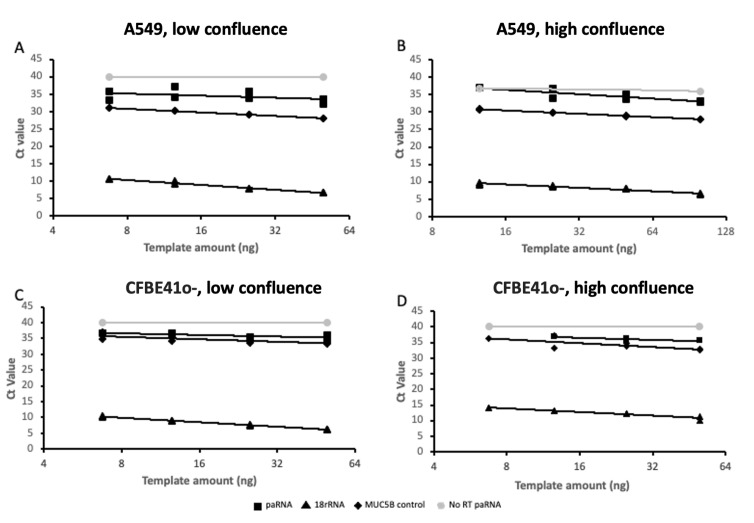
AC061979.1 expression validation in cell culture. Two-fold serial dilutions of A549 (**A**,**B**) and CFBE41o- (**C**,**D**) RNA extracts obtained from low- (<30%; (**A**,**C**)) and high- (>70%; (**B**,**D**)) confluence cells were analysed by probe hydrolysis RT-qPCR for 18S rRNA (triangle), MUC5B (diamond) and AC061979.1 pancRNA (square) expression across two-fold serial dilutions, with a no RT of AC061979.1 reaction set included as a negative control (grey circles). Data are expressed in log2-linear scale and are representative of three independent biological experiments and dual technical replicate Ct’s points shown.

**Figure 5 ncrna-08-00083-f005:**
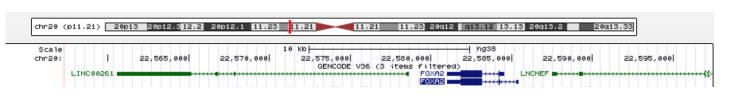
UCSC genome browser genomic location of the annotated lncRNA LINC00261 and FOXA2. LINC00261 and lncRNA neighboring enhancer of FOXA2 (LNCNEF)—green; FOXA2—dark blue.

## Data Availability

Publicly available datasets were analyzed in this study. The SRA/GSM accession numbers for each study can be found in Appendix A.

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
