# Peer review of "The Idiopathic Pulmonary Fibrosis-Associated Single Nucleotide Polymorphism RS35705950 Is Transcribed in a MUC5B Promoter Associated Long Non-Coding RNA (AC061979.1)"

_ncrna, 2022, doi:10.3390/ncrna8060083_

Round 1

Reviewer 1 Report

In the present original article Neatu et al. investigate the association of SNP RS35705950 with lncRNA involved in development of idiopathic pulmonary fibrosis.

The topic is of great importance for elucidating the role of lncRNA in regulation of gene expression and the development of different tissue pathology. Nevertheless, the present manuscript needs some improvements to be suitable for publication in ncRNA.

Figure 4 – Why authors present the data from RT-qPCR as Ct value and not using ΔΔCt method? Standard deviation and statistical analysis are not provided. The data from primary cell culture is not presented. The labeling of the graph should be improved for better understanding of which panel corresponds to which cell type.

Only data from adenocarcinoma, CF, and patients derived primary cells is obtained. What about healthy cell lines and donors?

Can authors provide data demonstrating how the expression of the protein codding genes, e.g. FOXA2, are changed?

Author Response

We thank the reviewer for their supportive comments and constructive ideas. 

  • We have elected to present the AC061979.1 RT-PCR data as delivered in Figure 4 to underscore the need for very high RNA template loading in RT-PCR for reliable detection. Presentation of the ΔCt values or ΔΔCt would not highlight the template abundance issue or validate the RNA-SEQ results by demonstrating low copy numbers. We nonetheless deliver ΔCt ±SD values in the main text body; ΔΔCt representation, in our opinion, is not appropriate in this specific context as no treatment or time-matched comparisons are performed with the A549 experiments (confluence is by nature neither a time-matched, nor treatment-related condition). In HAEpC experiments loss of AC061979.1 detection precludes ΔΔCt data expression.
  • Text indicating the statistical testing methodology used in this study has been added to the methods section, and we apologise for this oversight. Statistical significance is now reported in the manuscript where p > 0.05. 
  • Given the lack of AC061979.1 detection under some conditions in HAEpC assays, we thought outcome summation in ΔCt format in the text body would suffice. HAEpC experiments did not involve RNA serial dilutions in RT-PCR assays, but direct loading of the highest RNA amount possible based on the outcomes of the A549 experiments. Therefore text-only presentation is warranted. 
  • Figure 4 labelling has been updated as requested.
  • Cell line data are by nature 'unhealthy' since immortalisation or propagation in vitro requires either viral antigen transformation or involves tumor origin. The four donor samples used involve healthy resection tissue as well as transplant donor lung-derived cells as stated in the methods section; we therefore believe this addresses the reviewer's concerns. 
  • We have added data on the expression of FOXA2 as requested by the reviewer as a new supplementary dataset spanning both A549 cells and HAEpC's using RNA extract stocks from the original experiments. Unfortunately the CFBE41o- stocks were degraded precluding any analysis; we trust these results, which match expectations on FOXA2 underexpression in response to mucosecretion induced through IL-13 stimulation in HAEpC's, is adequate. 

Reviewer 2 Report

The study is relevant and of great interest in the IPF fiel. Perhaps the authors can take into account some suggestions:

1. Authors retrieved 27 independent databases from human alveolar and bronchial samples with different biological backgrounds (infected, immortalized, diseased), as enlisted in Supplementary Data. Was the sequencing quality comparable among all databases? If the biological context was not relevant, why not to include NGS databases from lung fibroblasts, specially if their sequencing depth allows to include transcript variant analysis (PMID: 24805851 , 31110176 and/or 31600171 )?

2. Improve Supplementary Table 1 (# of samples, sequencing depth, reads quality, RNA purification method used, levels of ncRNA within each dataset; mark which data are own data)

3. For Supplementary Table 2, does the predicted coding potencial differs among species?

4. Could your model explain the distal enhanced binding of FOXA2 downstream of the MUC5B SNP? By checking, for instance, FOXA2 motifs on the ncRNA? Are there motifs on the ncRNA for other transcription factors? Please enlist.

5. What are the exact genomic coordinates of pancRNA AC061979.1?

I hope my comments help to improve the quality of this manuscript and the relevance of their data.

Author Response

We thank the reviewer for their highly constructive comments which we hope we have addressed and implemented in full as detailed below. 

  • Sequencing quality between studies: We thank the reviewer for identifying this ommission in our data processing. Data with >Q20 bases were included in the analysis only, the methods section has been updated accordingly, and supplementary table 1 now indicates studies excluded from the analysis. 
  • Fibroblast datasets requested have been included in the analysis and demonstrate no reads within the region of interest. Figure 2 panel C has been added in the manuscript to illustrate this finding. 
  • Supplementary table 1 has now been updated to the reviewers' instructions indicating presence/absence of reads in the region of interest per dataset and IGV views of coverage. 
  • We have mapped the predicted binding sites for SMAD and FOXA2 due to their biological relevance in MUC5B expression regulation, as per reviewer's recommendation and created a new supplementary figure. This indicates a single FOXA2 binding site within exon 2 only, as previously presented by Helling et al 2017 and in the manuscript proximal to the SNP. There are two SMAD binding sites in the predicted intronic sequence, which could conceivably be affected on account of aberrant splicing. Text supporting these predicted binding sites has been added to the main body of the manuscript. 
  • The coordinates of the pancRNA were stated in paragraph 1, results section 3.1 (line no. 187). To improve reader experience the coordinates are now stated also in the legend of Figure 1 . We have attempted to validate this finding by 5' RACE using total RNA and RNA extracts enriched for pancRNA AC061979.1 using custom pulldown probes but unfortunately have been unable to retrieve >Q20 data to date. These results will be presented in a future manuscript since significant troubleshooting is required to solve low copy number RACE. 
  • We carefully considered the reviewer's suggestion regarding coding potential in AC061979.1. This is a valid question regarding its potential function, meriting of future exploration using proteomic approaches. However at this point in time, we feel that evaluation of coding potential might be premature since no data on transcript expression, other than locus conservation, is available beyond homo sapiens. For this reason we have refrained from including any analysis on coding and translation potential in this manuscript. However we will consider this work in future efforts on understanding the mechanistic involvement of AC061979.1 in MUC5B expression regulation. 

Round 2

Reviewer 1 Report

-

Reviewer 2 Report

Authors have addressed all comments by additional analysis, panels, data and/or valid discussions. From my point of view their revised manuscriptimproved its quality, with no doubt their data are of high relevance into the pneumology field. Therefore, I suggest acceptance of their revised work.